# Metabolites Facilitating Adaptation of Desert Cyanobacteria to Extremely Arid Environments

**DOI:** 10.3390/plants11233225

**Published:** 2022-11-24

**Authors:** Siarhei A. Dabravolski, Stanislav V. Isayenkov

**Affiliations:** 1Department of Biotechnology Engineering, Braude Academic College of Engineering, Snunit 51, Karmiel 2161002, Israel; 2Department of Plant Food Products and Biofortification, Institute of Food Biotechnology and Genomics, The National Academy of Sciences of Ukraine, Osipovskogo Str. 2a, 04123 Kyiv, Ukraine

**Keywords:** cyanobacteria, extracellular polysaccharides, compatible solutes, trehalose

## Abstract

Desert is one of the harshest environments on the planet, characterized by exposure to daily fluctuations of extreme conditions (such as high temperature, low nitrogen, low water, high salt, etc.). However, some cyanobacteria are able to live and flourish in such conditions, form communities, and facilitate survival of other organisms. Therefore, to ensure survival, desert cyanobacteria must develop sophisticated and comprehensive adaptation strategies to enhance their tolerance to multiple simultaneous stresses. In this review, we discuss the metabolic pathways used by desert cyanobacteria to adapt to extreme arid conditions. In particular, we focus on the extracellular polysaccharides and compatible solutes biosynthesis pathways and their evolution and special features. We also discuss the role of desert cyanobacteria in the improvement of soil properties and their ecological and environmental impact on soil communities. Finally, we summarize recent achievements in the application of desert cyanobacteria to prevent soil erosion and desertification.

## 1. Introduction

Life in the desert is constantly challenged by extreme abiotic stresses (such as high day/night temperature fluctuation, salinity, drought, excessive light, etc.), which negatively affects all desert lifeforms. Because of the geographical position and absence of vegetative coverage, most deserts are associated with rapid warming during daytime (up to 40–50 °C) and an abrupt drop-down of temperature to 0 °C at night. In addition, high solar radiation (up to 840 GJ km^2^ year^−1^) in the desert is another life-threatening factor [1]. The low water availability is another crucial abiotic stressor in desert ecosystems, where water could come from fog, groundwater, surface flow, atmospheric vapor and precipitation, and (rarely) rainfall. Furthermore, because of the high temperatures, surface-available water has high evaporation rates, which greatly decreases the time range of its potential use [2].

The desert soil type is defined as dry “Aridisols” with very low organic matter and nitrogen content, high phosphate, magnesium, and calcium carbonate content, and slightly alkaline pH because of the exposure to high-temperature fluctuation, strong wind erosion, water deficiency, and sedimentation [3].

The high salinity of desert soils represents another crucial problem for desert life. High Na^+^ content in soil restricts water conductance, soil porosity, and aeration. Because of the high osmotic stress and severe ion toxicity and imbalance, plants dwelling in the deserts are greatly limited. Thus, surplus amounts of chloride and sodium ions negatively affect plant protein metabolism, membrane fluidity, and enzyme functionality [4,5]. Furthermore, high soil salinity leads to additional water stress and nutritional imbalance in plants. This type of abiotic stress is usually accompanied by oxidative stress because of the generation of reactive oxygen species [5]. Also, high soil salinity has a negative impact on the microbial diversity of rhizospheres [6]. Thus, soil salinity greatly reduces the capacities of the plant to survive in saline and dry soils because of its direct impact on plant growth and development and indirect reduction of microbial community around roots associated with plant protection.

However, the dry desert soils are characterized by the presence of biological soil crust (BCS)–soil-surface communities that are composed of soil particles mixed with fungi, lichens, filamentous cyanobacteria, and mosses. The cohesive nature of BSCs allows them to maintain soil fertility, protect soil from wind and water erosion, prevent nutrient loss, and improve water holding capacity [7]. The diverse components of BSCs communities are known for their different contributions to nourishing the dry environments with nutrients. For example, some microbial components produce a polysaccharide sheath on the soil surface, and other microbes can fix nitrogen and carbon from the atmosphere, which are further metabolized by other fungi, and plants, thus increasing soil fertility and enhancing ecosystem stability [8]. It is known that extracellular polysaccharide (EPS)-producing bacteria that originated from stressed extreme environments can provide more efficient resistance mechanisms to abiotic stress compared to bacteria derived from non-stress conditions [9]. Several recent publications in which diverse genomics, proteomics, and metabolomics techniques were used suggest that cyanobacteria originated from extremely dry environments and possess a set of unique features, allowing them to survive and form prosperous communities with other taxa (algae, fungi, mosses, and other bacteria) [10,11,12].

Therefore, in this review, we focus on the cyanobacteria isolated from extreme arid ecosystems and deserts, their metabolic mechanisms to cope with drought stress, and their potential application in restoring degraded soil and preventing the spread of desertification.

## 2. Cyanobacteria as Pioneers Forming the Core Community

The ability of cyanobacteria to cope with different environmental stresses allows them to be pioneers in the colonization of terrestrial ecosystems. During colonization, cyanobacteria protect soil from erosion, improve texture, promote carbon and nitrogen fixation, increase soil organic matter, and closely cooperate with other organisms (bacteria, algae, archaea, fungi) to facilitate subsequent growth of lichens, mosses, and vascular plants. In natural conditions, filamentous cyanobacteria initiate colonization of the top millimeters of the soil and form BCSs by assembling second-line colonization species [13].

Recent research has demonstrated that the bundle-forming, filamentous, non-nitrogen-fixing cyanobacterium *Microcoleus vaginatus* was the main source of organic carbon and the dominant member and spatial organizer of the biocrust microbiome in the Chihuahuan Desert and the Great Basin Desert. In particular, *M. vaginatus* formed a specialized cyanosphere shape enriched in copiotrophs and diazotrophs species with significantly abundant nitrogen-fixation genes. Some species have segregated away from the *M. vaginatus* cyanosphere through competition for light and CO_2_ and because of the gradually reducing organic carbon concentrations [14].

Investigation of the edaphic and lithic microhabitats in the Atacama Desert has shown that the diversity of cyanobacteria declines as precipitation levels decrease. Among 21 identified cyanobacteria species, some were shared across most collection sites (*Scytonema hyalinum*, *Nostoc* sp., *Trichocoleus sociatus*, *Pseudophormidium* sp., and *P. minor*), and some were unique for lithic (*Alteriella* sp., *Gloeocapsopsis* sp. and *Kastovskya*) and edaphic (*Phormidesmis* sp., *Nodosolinea epilithica*, and *Mojavia pulchra*) microhabitats. These results suggest that the composition of both edaphic and lithic communities strongly correlates with the level of xeric stress [15]; therefore, the most suitable strain could be selected for every soil type/available water. Similarly, differences were shown for the Sahara Desert cyanobacteria, where *Microcoleus* sp. dominated in the less saline site, while the more saline sites had a high abundance of heterocystous cyanobacteria and the filamentous non-heterocystous *Pseudophormidium* sp. and the unicellular cf. *Acaryochloris*. Other identified cyanobacteria (*Microcoleus steenstrupii*, *Microcoleus vaginatus*, *Scytonema hyalinum*, *Tolypothrix distorta*, and *Calothrix* sp.) were also found in other ecosystems with less severe environmental conditions [16].

Interestingly, a recent global investigation demonstrated the relative stability of soil cyanobacteria community composition, abundance, and species richness during soil development (over centuries and millennia across contrasting ecosystems worldwide). However, change in vegetation types (grasslands, shrublands, forests, and croplands) clearly affects the composition of soil cyanobacteria communities. Consequently, regardless of soil age, grasslands and shrublands were dominated by photosynthetic cyanobacteria, while forests were dominated by non-photosynthetic cyanobacteria (Vampirovibrionia). These results highlight the major role of vegetation in the modulation of cyanobacteria abundance, diversity, composition, and nutrition type (nonphotosynthetic and photosynthetic) during soil development [17].

## 3. Ecological Significance of Soil Consortia Dominated by Cyanobacteria

The close association between cyanobacteria and microalgae with other aerobic or anaerobic microorganisms is known as a consortium, as either mutualism or parasitism [18]. Under mutualism, microalgae and bacteria exchange micronutrients, macronutrients, and hormones [19]. In parasitism, some heterotrophic bacteria secrete different enzymes (such as cellulases, glucosidases, chitinases, and others) to suppress growth and degrade other members of the consortium to acquire nutrients [20].

Recently identified structure, diversity, and co-occurrence patterns of hypolithic communities from the Namib Desert showed the dominance of photosynthetic cyanobacteria of the orders Oscillatoriales, Pseudanabaenales, and Synechococcales. Interestingly, the calculated co-occurrence network showed a highly modular structure with almost exclusively positive co-occurrences between Cyanobacteria and Alphaproteobacteria taxa. Further, members of the Alphaproteobacteria class were the most common connectors, interacting simultaneously with different active heterotrophic and phototrophic community members. This evidence implicates both Cyanobacteria and Alphaproteobacteria as keystone taxa, coordinating biogeochemical cycling, community structure, and organization in these niches [21].

An investigation of the global diversity of desert hypolithic cyanobacteria showed the dominance of Phormidium and Chroococcidiopsis taxa. Interestingly, no nitrogen-fixing cyanobacteria have been identified; instead, all nitrogenase sequences were affiliated with the Proteobacteria (orders Burkholderiales, Rhizobiales, and Rhodospirillales) [22]. Similarly, Hypolithic mosses in the Mojave Desert (such as *Syntrichia caninervis*, *Tortula inermis*, and *Bryum argenteum*) co-occurred with cyanobecteria, but also acquired nitrogen from non-cyanobacterial diazotrophs, generally from Proteobacteria taxa members [23]. Desiccation tolerant cyanobacteria *Nostoc flagelliforme* was shown to closely associate in communities with diverse Actinobacteria, Proteobacteria, Acidobacteria, and Bacteroidetes taxa [24]. During rehydration, *N. flagelliforme* secretes over 200 various extra-cellular hydrolytic enzymes and membrane transport proteins to communicate with accompanying epiphytic bacteria and acquire required nutrients [25].

Therefore, cyanobacteria as a dominant species group in the consortium orchestrate complex exchanges of nutrients and signaling molecules with neighboring bacteria, algae, and mosses to facilitate stable growth and stability of the entire ecosystem.

## 4. Acclimation Strategies of Cyanobacteria to Survive Extreme Arid Conditions

General strategies used by cyanobacteria to survive desiccation/rehydration cycles in xeric environments are well-known: production of EPS, recruiting chaperones to maintain protein integrity, up-regulation of DNA repair and the oxidative stress protection system, synthesis of compatible solutes and ion channels to adapt to low levels of available water, etc. [26,27]. Further, we focus specifically on cyanobacteria isolated from a desert environment, their EPS, and closely related compatible solutes synthesis systems as the main protective mechanisms. In addition, we discuss the specific adaptation features that help them to survive in extreme water deficiency conditions and distinguish sugars and EPS production and composition in desert cyanobacteria from general pathways that are known for other cyanobacteria.

### 4.1. Biosynthesis and Role of EPSs

EPSs compose a significant part of the extracellular matrix and up to 95% of bacterial weight. Besides polysaccharides, they may consist of nucleic acids, proteins, and lipids [28]. While they have two basic forms (slime and capsular), EPSs are further classified as neutral, polycationic, and polyionic (the majority of EPSs) [29]. EPSs modulate hydrological properties of soil and retain water, thus slowing the desiccation rate and protecting the photosynthetic apparatus [30,31]. Such protection is vital for other organisms in the BSC, such as green algae (*Chlorella* sp.), which could not revive even after slow desiccation in the absence of the cyanobacteria [32]. In addition, cyanobacterial EPS may also constitute a valuable source of carbon for heterotrophic microorganisms [33]. Apparently, EPSs synthesis is a slow process that could not quickly respond to fast-occurring dehydration and desiccation. However, during slow and gradual desiccation, cells have time to prepare for dehydration via up- or down-regulation of the necessary network of genes to facilitate quick revival. Also, during this preparation step, RNAs that are involved in some crucial functions (such as photosynthesis) are stabilized/protected, possibly by intrinsically disordered proteins, thus providing prompt resumption upon rewetting [34]. Indeed, as was shown for *Nostoc flagelliforme*, an increase in EPS content and ROS level was higher after 6 h of dehydration compared to 4 h of rehydration. Interestingly, SOD and POD activities were decreased under dehydration, while CAT was enhanced. These data suggest that dehydration could induce an increase in EPS contents and the ROS level. Consequently, the dysfunction of the photosynthesis process, as the main ROS-production site, could have resulted from significant ROS accumulation [35].

A filamentous terrestrial cyanobacterium *N. flagelliforme* synthesizes the ultraviolet (UV)-screening pigments, such as scytonemin (SCY) and mycosporine-like amino acids (MAAs), located in the EPS to protect the photosynthesis apparatus from damaging UV radiation. A partial SCY loss (~3.7%) resulted in the reduction of structural stability of the EPS matrix and slower photosystem II activity recovery after desiccation [36]. Experiments on endolithobiontic cyanobacteria Halothece sp. from Atacama Desert showed that UV radiation regulates *scyB* (the main gene that is responsible for the scytonemin biosynthesis) transcription and scytonemin biosynthesis. Under low humidity and UV-A radiation, the ratio scytonemin/chlorophyll a and the transcription of *scyB* gene increased to a maximal 1.7-fold value [37]. Further, the SCY functions in heat dissipation from absorbed UV radiation, antioxidant activity, and sunscreen ability, thus suggesting its crucial role in protecting cyanobacteria surviving under environmental stresses [38,39].

The comparison of the *mysABCD* genes (which are responsible for MAA biosynthesis) from *Nostoc verrucosum* (sensitive to desiccation) and *Nostoc commune* (tolerant to extreme desiccation) demonstrated that *N. verrucosum* misses the *mysD* gene in the cluster, but it locates in the 3`downstream region with an anti-parallel orientation. Further, *N. verrucosum* MAAs comprise over 90% of porphyra-334, an atypical cyanobacterial MAA, which indicated little or no radical scavenging activity in vitro. On the contrary, N. commune produced the glycosylated derivatives of porphyra-334, which are potent radical scavengers, thus enhancing adaptation to adverse environmental conditions [40].

The recent investigation of 95 cyanobacterial genomes demonstrated that *mycC* (encoding ATP-grasp ligase) duplication within the myc cluster is strictly limited to drought-tolerant cyanobacteria. In contrast to canonical *mysCs*, *mysC2* strictly condenses the α-amino group of mycosporine-ornithine to another 4-deoxygadusol, while *mysC3* catalyzes the linkage of the δ- or ε-amino group of ornithine/lysine to 4-deoxygadusol, yielding mycosporine–ornithine or mycosporine–lysine, respectively (Figure 1). Therefore, ATP-grasp ligase duplication represents a new adaptation of MAAs biosynthesis pathway to increase cyanobacteria tolerance for UV in drought conditions via the production of complicated MAAs with multiple chromophores [41].

In total, the MAAs biosynthesis pathway represents a very adaptive mechanism to produce a diverse set of UV-protecting pigments in different organisms (such as cyanobacteria, corals, and red algae). Recently, an additional enzyme named *mysH* was identified in *Nostoc lincki* that is responsible for further shinorine and porphyra-334 conversion to palythines [42], suggesting high potential for the development of novel sunscreens with required properties.

The structural dynamics of the EPS matrix upon expansion and shrinkage in *N. flagelliforme* are regulated with *wspA*, an abundantly secreted β-galactosidase, in which transcription is moisture-dependent. WspA protein was characterized by glycoside hydrolysis and transgalactosylation activities, which facilitate softening or thickening of the EPS matrix under moistening or drying processes, respectively. Such flexible coordination through moisture availability of desiccation–rehydration cycles allows the cyanobacterial EPS matrix to balance cell growth and stress resistance [43]. Additionally, the EPS matrix has a strong and resilient pH buffering property, protecting cells from low pH, which is often seen in dryland rainwater polluted with nitrogen or sulphur [44]. Indeed, a recent study demonstrated that a cultured strain of *N. sphaericum* produced massive EPS but was sensitive to desiccation, confirming the idea that the amount of EPS and its production speed do not provide desiccation tolerance. The analysis of EPS from a desiccation-tolerant N. commune indicated that conservative *sodF* (superoxide dismutase) and highly diverse *wspA* genes are crucial for proper environmental adaptation and survival [45]. As was further revealed by SNP (single nucleotide polymorphism) and phylogenetic analysis of the *wspA* gene and WspA protein in *N. flagelliforme*, existing plentiful nucleotide variations may be related to different regions and adaptation to dryland conditions [46].

### 4.2. Compatible Solutes

Compatible solutes are low-molecular-mass organic compounds that are known to protect against osmotic imbalance, facilitate reduction of proteins unfolding and denaturation, and stabilize other macro-molecules under low water potential. In cyanobacteria, the main compatible solutes are sucrose, glycine betaine, glucosylglycerol, and trehalose, which have a long history of investigation (reviewed in [47]). In contrast with higher plants, proline and glucosyl glycerate represent less common (or secondary) compatible solutes that have been identified in some cyanobacteria [48], while glycogen and PHB (polyhydroxybutyrate) are considered as multifunctional stress-related molecules with a wide physiological role [49].

Recent phylogenetic and comparative genomic investigation of 650 cyanobacterial genomes suggested that the expansion of gene families associated with sucrose and trehalose biosynthesis is a common desiccation-protective strategy for terrestrial cyanobacteria to cope with extreme environmental conditions. In particular, the treZY cluster (encoding malto-oligosyltrehalose trehalohydrolase and synthase) and the sucrose synthase gene were significantly enriched in terrestrial strains [12] (Figure 2). The importance of trehalose metabolism for desiccation tolerance was shown in the comparison of desiccation-tolerant and sensitive strains of *Leptolyngbya ohadii*. Desiccation tolerant strains have a treZY cluster and a long version (1127 aa) of *TreS* (encoding trehalose synthase), while desiccation-sensitive cyanobacteria have a short (546 amino acids) version. Supposedly, the maltogenic amylase and maltokinase domains that are located on the C-terminus in the longer *TreS* version and involved in the degradation of glycogen to produce trehalos, are most likely responsible for trehalose-related desiccation tolerance. However, both *TreS* versions (short and long) are upregulated in *L. ohadii* during desiccation [10]. Further research has shown that the addition of trehalose before dehydration significantly improved cell recovery. Further, genes responsible for sucrose biosynthesis were far less affected by desiccation in comparison with trehalose-related pathways, suggesting the primary role of sucrose as a compatible solute in *L. ohadii* [34].

Interestingly, *TreS* and *TreH* (neutral trehalase) genes were identified in the genome of desiccation-tolerant extremophile Gloeocapsopsis sp. UTEX B3054, while no homologous genes of *treZ* or *treY* were found [11]. A closely related and extremely desiccation-tolerant cyanobacterium *Gloeocapsopsis* AAB1, isolated from the Atacama Desert, was shown to synthesize under desiccation conditions between 40–80 times more sucrose and about 5–15 times more trehalose than the other cyanobacteria [50]. Similarly, *Desmonostoc salinum* CCM-UFV059, a salt and desiccation-stress-tolerant cyanobacterial strain isolated from a saline–alkaline lake, mostly over-accumulated high amounts of sucrose and, to a lower extent, trehalose under desiccation conditions [51].

It is important to note that trehalose is one of the oldest studied of the compatible solutes that are are widely present in archaea, bacteria, fungi, algae, and plants [52,53], with other pathways that are used for its synthesis (such as TreP and TreT-based, and others) (Figure 2). Recent research demonstrated the involvement of trehalose in salt- and water-deprivation stresses in archaea, yeast, and plant-pathogenic fungi [54,55,56]. Also, for the first time, the trehalose biosynthesis under desiccation stress was identified in the marine red algae *Porphyra umbilicalis* [57], because generally trehalose is considered a compatible solute for terrestrial organisms. Interestingly, instead of direct cellular protection, in plants, trehalose plays more of a signaling and regulatory role, mediating communication and interaction with symbiotic microorganisms in the rhizosphere [58].

These results show that EPS components and compatible solutes play a crucial role in cyanobacterial desiccation tolerance. However, the exact molecular mechanism regulating the rapid synthesis and composition of MAAs, SCY, and compatible solutes upon dehydration is still poorly understood. Because desert cyanobacteria represent some unique features of genome evolutional and adaptation to extreme arid conditions, further investigation, specifically in this ecological group, is required (Figure 3). Such cyanobacteria could effectively improve soil properties and open new frontiers in industrial biotechnology.

## 5. Use of Desert Cyanobacteria for Soil Remediation

There are many modern sustainable engineered systems based on the application of cyanobacteria in combination with other organisms for different environmental and bio-technological purposes [59]. Although various cyanobacteria species are widely used in industry and agriculture, there are no examples of commercial use of desert cyanobacteria as PGPB or mixtures to improve agricultural soil properties [60]. However, desert cyanobacteria are widely studied as promising tools for desert soil restoration and related wastewater remediation [61].

### 5.1. Cultivation of Desert Cyanobacteria in Wastewater

Cultivating sand-consolidating cyanobacteria using wastewater has a significant advantage compared to other cultivation methods, because it does double work with water remediation and effectively supports the growth of cyanobacteria in desert areas. Currently, the cultivation of *Scytonema* sp. is widely accepted as the most promising species for industrial use. For example, recent research demonstrated that sand-consolidating cyanobacteria *Scytonema hyalinum* cultivated in the municipal wastewater effectively accumulated biomass and removed nutrients from wastewater at 20–30 °C [62]. Similarly, desert cyanobacterium *Scytonema javanicum* gradually removed nutrients and accumulated biomass during cultivation in artificial synthetic wastewater [63].

Recently, the growth of *Scytonema javanicum* was compared in undistilled/distilled wastewater and an artificial culture medium. While wastewater reduced the chlorophyll concentration, it did not affect the biomass accumulation and photosynthetic recovery after long-term storage [64]. Similar to *S. hyalinum*, the optimal cultivation temperature for *S. javanicum* was defined at 25–30 °C, which resulted in increased activity of photosynthetic system II, accelerated biomass accumulation, and improved nutrient removal from wastewater. Higher temperatures caused damage to the photosynthetic apparatus and decreased final biomass accumulation [65].

In total, these results provide scientific evidence for the development of effective modern technology to solve both industrial and ecological problems with wastewater remediation and desert soil improvement.

### 5.2. Desert Cyanobacteria for Soil Restoration

The effectiveness of cyanobacteria-based approaches to restore degraded soil, prevent the spread of desertification, and/or increase productivity relies on a deep knowledge of the native and dominant cyanobacteria community of these ecosystems. Unfortunately, many cyanobacterial biocrust communities remain unknown or poorly characterized in various regions of the world [66]. However, many studies have reported the successful application of cyanobacteria to improve soil indicators.

Recently, a biofilm-based *Microcoleus vaginatus* cultivation method on the most readily available and inexpensive shifting sand as an attached substrate was studied. The maximal photosynthetic activity and cyanobacteria biofilm biomass productivity was achieved under controlled water content at 10% after 15–25 days of cultivation, resulting in a stable and easily peeled biofilm formed through filamentous binding and exopolysaccharide cementing [67].

Interestingly, the temperature was reported as the most important controlling factor for the growth and productivity of two filamentous cyanobacteria *Scytonema tolypothrichoides* and *Tolypothrix bouteillei*, known as effective soil stabilizers in arid areas because of the production of significant amounts of EPS. The optimum growth range for *S. tolypothrichoides* and *T. bouteillei* was achieved at 27–34 °C and 22–32 °C, respectively. The temperature required for maximum EPS production for *S. tolypothrichoides* and *T. bouteillei* was nearly identical 28–34 °C and 27–34 °C, respectively [68].

Recently, features of cyanobacterial EPS synthesis and release were evaluated for three common biocrust-forming cyanobacteria (*Phormidium ambiguum*, *Scytonema javanicum*, and *Nostoc commune*) in controlled laboratory conditions in liquid media and on sandy soil microcosms. Despite the greatest growth and EPS release demonstrated by *P. ambiguum*, *S. javanicum* demonstrated the highest growth and highest soluble EPS content on the sandy soil. Surprisingly, *N. commune* showed no significant growth after its inoculation on the sandy soil. The content of condensed soil EPS fractions was similar for both *P. ambiguum* and *S. javanicum*. These results indicate that features of EPS (released in liquid culture, soluble, and condensed in sandy soil microcosms) should be evaluated to select suitable strain for large-scale cyanobacteria applications in soil restoration [69].

An interesting method to enhance soil fertility and prevent erosion with inoculation of pelletized cyanobacteria from desert soil biocrusts was suggested. Fresh cultures of two N-fixing genera (Nostoc and Scytonema) and a non-heterocystous filamentous genus (Leptolyngbya) were combined with bentonite powder and sand and extruded into pellets. Greenhouse experiments on soils from three arid regions in Australia demonstrated that pellets can dissolve completely and spread out in all treatments. However, Scytonema and the consortium of the three cyanobacteria species showed higher chlorophyll contents and lower albedo compared to the Nostoc and Leptolyngbya inoculations. These results provide a new and effective method of cyanobacteria delivery and degraded soil inoculation with suitable cyanobacteria species [70]. Another report suggests using the metallorganic framework and carboxymethyl cellulose to create network-structured nanocomposite material to retain eutrophic water containing aquatic cyanobacteria and nutrients. Inoculation of this combination in the soil promotes growth and biocrusts formation of desert cyanobacteria, subsequently inhibiting desertification [71].

Interestingly, numerous studies have suggested that terrestrial desert conditions have similar features to Mars, such that desert regions on Earth provide a model for the extreme aridity of the Martian surface [72]. The first attempts were based on the inoculation of desert soil with cyanobacteria to form crusts and stabilize soil particles [73]. More recently such experiments have been transferred to the ISS (International Space Station), where survival of dried Chroococcidiopsis cells was confirmed after exposure to UV radiation for 469 days to a Mars-like atmosphere for 722 days [74] and to space vacuum and solar radiation [75]. As it was recently shown, the desert cyanobacterium *Chroococcidiopsis* sp. CCMEE 029 had no significant genomic alterations compared to the reference genome after 1.5 years of exposure to cosmic ionizing radiation and Mars-like conditions outside of the ISS [76].

## 6. Conclusions

Based on the above evidence, we suggest that the ability of desert cyanobacteria to survive in extremely dry environments and colonize harsh environments including desert sand facilitates and provides more favourable conditions for the later colonization of land by plants and other types of living organisms. Future investigation of molecular and generic factors responsible for regulation and production of bioactive components of the EPS (such as MAAs, SCY, and compatible solutes) would promote wider agricultural and industrial application of specialized cyanobacterial strains. As a pioneer of frontier habitats, cyanobacteria have a great unexplored potential for future application to prevent and reverse desertification, directly improve soil properties, and increase productivity of agriculture in arid zones (Figure 4).

## Figures and Tables

**Figure 1 plants-11-03225-f001:**
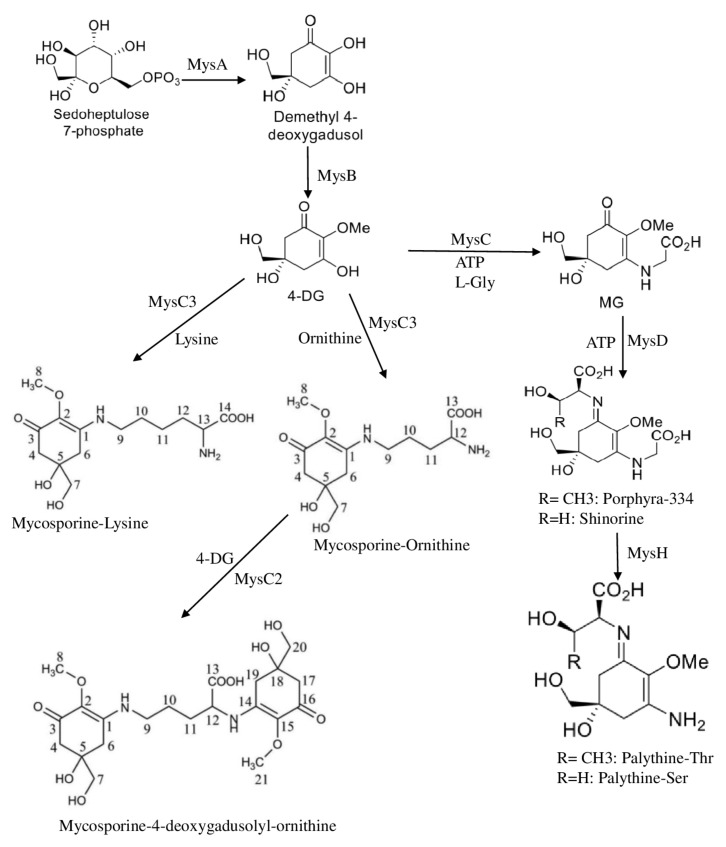
Biosynthesis pathway for the MAAs. 4-DG (4-deoxygadusol), MG (mycosporine–glycine). *mysA* (dimethyl 4-degadusol synthase), *mysB* (O-methyltransferase), *mysC* (ATP-grasp ligase), *mysD* (D-Ala–D-Ala ligase), *mysH* (dehydrogenase/reductase and a nonheme iron(II)- and 2-oxoglutarate-dependent oxygenase).

**Figure 2 plants-11-03225-f002:**
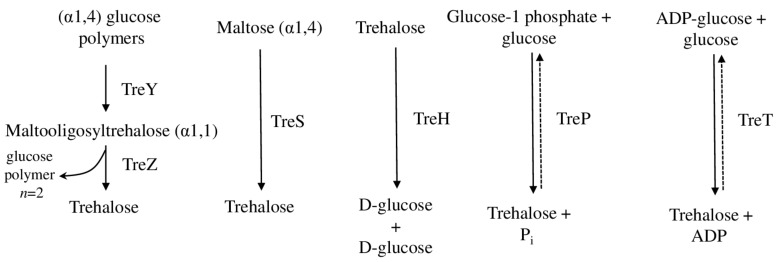
The trehalose biosynthetic pathways. *TreY* (maltooligosyl–trehalose synthase), *TreZ* (maltooligosyl–trehalose trehalohydrolase), *TreS* (trehatose synthase), *TreH* (trehalase), *TreP* (trehalose phosphorylase), *TreT* (trehalose glycosyl-transferring synthase).

**Figure 3 plants-11-03225-f003:**
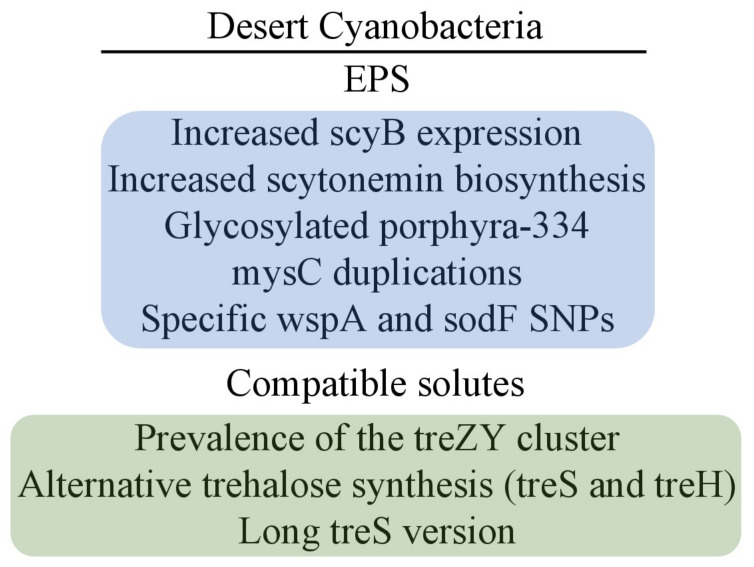
The main EPS and compatible solute features that distinguish desiccation-tolerant cyanobacteria from desiccation-sensitive species.

**Figure 4 plants-11-03225-f004:**
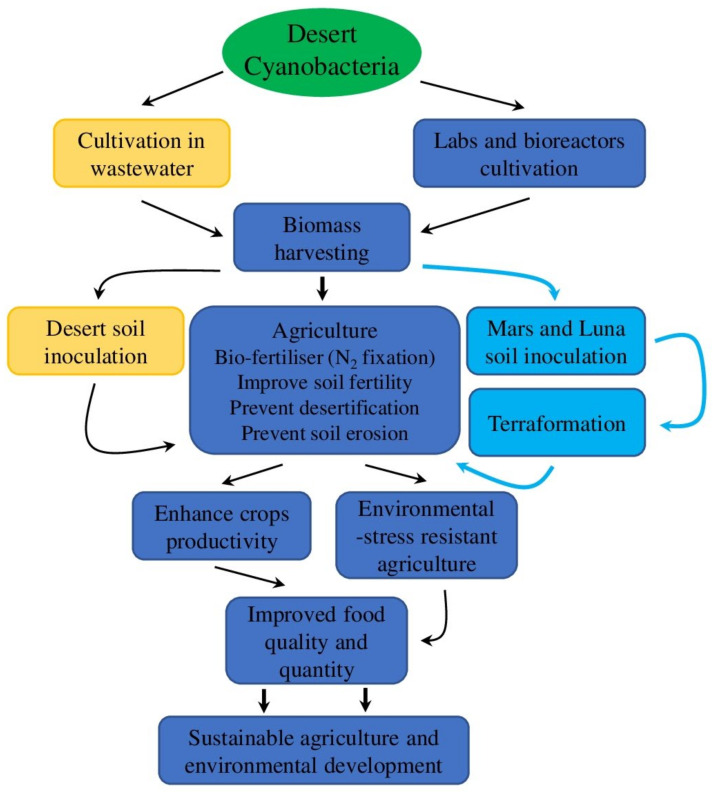
A hypothetical scheme exhibiting the potential roles of desert cyanobacteria. Potential application of desert cyanobacteria to solve ecological problems is shown in orange; future space exploration areas are shown in cyan; convenient ways to use desert cyanobacteria for sustainable agriculture and environmental management are shown in blue.

## Data Availability

Not applicable.

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
