# Peer review of "Metabolites Facilitating Adaptation of Desert Cyanobacteria to Extremely Arid Environments"

_plants, 2022, doi:10.3390/plants11233225_

Round 1
Reviewer 1 Report
The topic of manuscript is interesting for the Journal and the manuscript quite well written even if it needs to be checked for English lenguage. It is presented in context with updated literature.
Author Response
Dear Editor and Reviewers,
We greatly appreciate your critical evaluation of our manuscript and helpful comments. Our reply to your comments would be provided point by point, where “A” stands for “Authors”, and “L” for “Lines”, where changes have been implemented. The language of the entire manuscript has been checked and corrected.
____________________________________________________________________________
The topic of manuscript is interesting for the Journal and the manuscript quite well written even if it needs to be checked for English lenguage. It is presented in context with updated literature.
A: The language of the entire manuscript has been edited.

Reviewer 2 Report
This is a well-written review emphasizing the crucial role of cyanoprocaryotes in extreme habitats, such as deserts, in the formation of biological soil crusts, their mechanisms to cope with the multiple stresses offered by harsh conditions, especially the extracellular polysaccharides and compatible solutes, as well as their further possible applications.
Author Response
Dear Editor and Reviewers,
We greatly appreciate your critical evaluation of our manuscript and helpful comments. Our reply to your comments would be provided point by point, where “A” stands for “Authors”, and “L” for “Lines”, where changes have been implemented. The language of the entire manuscript has been checked and corrected.
____________________________________________________________________________
This is a well-written review emphasizing the crucial role of cyanoprocaryotes in extreme habitats, such as deserts, in the formation of biological soil crusts, their mechanisms to cope with the multiple stresses offered by harsh conditions, especially the extracellular polysaccharides and compatible solutes, as well as their further possible applications.
A: The language of the entire manuscript has been edited.

Reviewer 3 Report
The article “Metabolic adaptation of desert cyanobacteria to the extremely arid environments” presents a subject of plant research relevance and explores the literature of the area. The title and subject of the manuscript are very interesting from the methodological and practical point of view, suitable and adequate the abstract of the paper is factual concrete, realistic, understandable, self-readable. However the manuscript should be improve.
- The title of current manuscript is not matching to the manuscript information. Author should revise the title or add relative information on how cyanobacteria adaptation Metabolite or their role in metabolite production and modulation.
- The abstract need to be improve, the author should provide more reliable conclusion and outcome of their study.
- The current introduction did not provide a flow and easy understanding to read. Its look like the different paragraphs are just combine together with different information. The author should improve the introduction and connect each information to next paragraph.
- The objective of the current study is not clearly define. Dose the author clam to isolate cyanobacteria from the deserts for the experimental propose. As it a review article so author should state the objective relative to problem, role and important of cyanobacteria.
- Overall the manuscript is well arranged and well written.
- The figure 3 did not provide efficient information regarding the current statement, author should improve the figure to explant more about the pathway of EPS and interaction with metabolites in cyanobacteria.
- In conclusion author should state how this review can be helpful for future research and the important of cyanobacteria to mitigate stress
- The figure quality is so poor, author should provide more clearly quality of all figures.

Author Response
Dear Editor and Reviewers,
We greatly appreciate your critical evaluation of our manuscript and helpful comments. Our reply to your comments would be provided point by point, where “A” stands for “Authors”, and “L” for “Lines”, where changes have been implemented. The language of the entire manuscript has been checked and corrected.
____________________________________________________________________________
The article “Metabolic adaptation of desert cyanobacteria to the extremely arid environments” presents a subject of plant research relevance and explores the literature of the area. The title and subject of the manuscript are very interesting from the methodological and practical point of view, suitable and adequate the abstract of the paper is factual concrete, realistic, understandable, self-readable. However the manuscript should be improve.
- The title of current manuscript is not matching to the manuscript information. Author should revise the title or add relative information on how cyanobacteria adaptation Metabolite or their role in metabolite production and modulation.
A: The title was modified (L2-3) - The abstract need to be improve, the author should provide more reliable conclusion and outcome of their study.
A: As a review paper, there are no clear conclusions or outcomes. Several goals have been set (to overview the metabolic pathways, environmental impact, practical application, etc,) and discussed (please, see sections 1 to 5). The conclusion section (6th) is based only on discussed results. Please, keep in mind that no experiments were conducted by the authors – it is not required for the review paper. - The current introduction did not provide a flow and easy understanding to read. Its look like the different paragraphs are just combine together with different information. The author should improve the introduction and connect each information to next paragraph.
A: The introduction section was modified - The objective of the current study is not clearly define. Dose the author clam to isolate cyanobacteria from the deserts for the experimental propose. As it a review article so author should state the objective relative to problem, role and important of cyanobacteria.
A: As a review paper, there is no experimental part. The relevant (desert habitats) importance of cyanobacteria is discussed in section 2 - Overall the manuscript is well arranged and well written.
A: Thank you for your valuable opinion. - The figure 3 did not provide efficient information regarding the current statement, author should improve the figure to explant more about the pathway of EPS and interaction with metabolites in cyanobacteria.
A: Figure 3 is dedicated to the genetic and molecular features, which could help to distinguish desiccation-tolerant and desiccation-sensitive species of cyanobacteria, so It would be easier to classify and modify different species/strains.
“the pathway of EPS” – if the biosynthesis was meant, please, see the cited papers [28, 29]. The detailed analysis and discussion of the EPS biosynthesis process is beyond the scope of this focused review.
Otherwise, please, clarify/modify your comment. - In conclusion author should state how this review can be helpful for future research and the important of cyanobacteria to mitigate stress.
A: The conclusion section was modified (L406-409)
Otherwise, please, clarify your comment, in the current context it is not clear what kind of reply is required for this part of the comment: “the important of cyanobacteria to mitigate stress”
- The figure quality is so poor, author should provide more clearly quality of all figures.
A: The quality of all figures was re-checked (set for 300 DPI). Otherwise, please, clarify/modify your comment.
